# GROKKING IN LINEAR ESTIMATORS – A SOLVABLE MODEL THAT GROKS WITHOUT UNDERSTANDING

**Noam Levi**[†‡]**, Alon Beck**[†] **& Yohai Bar Sinai**[†]
[†]Raymond and Beverly Sackler School of Physics and Astronomy
Tel-Aviv University
Tel-Aviv 69978, Israel
[‡]École Polytechnique Fédérale de Lausanne (EPFL)
Switzerland
{noam@tau.ac.il, alonbk2@gmail.com, ybarsinai@gmail.com}

## ABSTRACT

Grokking is the intriguing phenomenon where a model learns to generalize long after it has fit the training data. We show both analytically and numerically that grokking can surprisingly occur in linear networks performing linear tasks in a simple teacher-student setup with Gaussian inputs. In this setting, the full training dynamics is derived in terms of the training and generalization data covariance matrix. We present exact predictions on how the grokking time depends on input and output dimensionality, train sample size, regularization, and network initialization. We demonstrate that the sharp increase in generalization accuracy may not imply a transition from "memorization" to "understanding", but can simply be an artifact of the accuracy measure. We provide empirical verification for our calculations, along with preliminary results indicating that some predictions also hold for deeper networks, with non-linear activations.

## 1 INTRODUCTION

Understanding the underlying correlations in complex datasets is the main challenge of statistical learning. Assuming that training and generalization data are drawn from a similar distribution, the discrepancy between training and generalization metrics quantifies how well a model extracts meaningful features from the training data, and what portion of its reasoning is based on idiosyncrasies in the training data. Traditionally, one would expect that once a neural network (NN) training converges to a low loss value, the generalization error should either plateau, for good models, or deteriorate for models that overfit.

Surprisingly, Power et al. (2022) found that a shallow transformer trained on algorithmic datasets features drastically different dynamics. The network first overfits the training data, achieving low and stable training loss with high generalization error for an extended period, then suddenly and rapidly transitions to a perfect generalization phase.

This counter-intuitive phenomenon, dubbed *grokking*, has recently garnered much attention and many underlying mechanisms have been proposed as possible explanations. These include the difficulty of representation learning (Liu et al., 2022), the scale of parameters at initialization (Liu et al., 2023), spikes in loss ("slingshots") (Thilak et al., 2022), random walks among optimal solutions (Millidge, 2022), and the simplicity of the generalising solution (Nanda et al., 2023, Appendix E).

In this paper we take a different approach, leveraging the simplest possible models that still display grokking-linear estimators. Due to their simplicity, this class of models offers analytically tractable dynamics, allowing a derivation of exact predictions for grokking, and a clear interpretation that is corroborated empirically. Our main contributions are:

- We solve analytically the gradient-flow training dynamics in a linear teacher-student $(T, S \in \mathbb{R}^{d_{\text{in}} \times d_{\text{out}}})$ model performing MSE classification. In this setting, the training and generalization losses $\mathcal{L}_{\text{tr}}, \mathcal{L}_{\text{gen}}$, are simply given by $||T - S||^2_{\Sigma}$, where the norm is

defined with respect to the training/generalization Gram matrices, $\Sigma_{\mathrm{tr}}$ and $\Sigma_{\mathrm{gen}}$ respectively. These matrices can be modeled with classical Random Matrix Theory (RMT) techniques.

- Grokking in this setting does not imply any "interesting" generalization behavior, but rather the simple fact that the generalization loss decays slower than the training loss, because the gradients are set by the latter. The grokking time is mainly determined by a single parameter, the ratio between input dimension and number of training samples $\lambda = d_{\mathrm{in}}/N_{\mathrm{tr}}$.

- Standard variations are included in the analysis:
    - The effect of different weight initializations is to generate an artificial rescaling of the training and generalization losses, increasing the effective accuracy value required for saturation and therefore increasing grokking time.
    - For small $d_{\mathrm{out}}$, Grokking time increases with output dimension due to effectively slower dynamics. This happens up to a critical dimension after which the measure of accuracy becomes insensitive to the value of the loss, reducing the grokking time.
    - $L_2$ regularization suppresses grokking in overparameterized networks as expected, while having a subtle effect on the grokking time in underparameterized settings.

- We further show semi-analytically that our results extend to architectures beyond shallow linear networks, including one hidden layer, with both linear and some nonlinear activations.

## 2  RELATED WORK

**Grokking**  Many works have attempted to explain the underlying mechanism responsible for grokking, since its discovery by Power et al. (2022). Some works suggest "slingshots" (Thilak et al., 2022) or "oscillations" (Notsawo et al., 2023) underlie grokking, but our explanation applies even without these dynamics. Other works identify ingredients for grokking (Davies et al., 2023; Nanda et al., 2023), analyze the trigonometric algorithms networks learn after grokking (Nanda et al., 2023; Chughtai et al., 2023; Merrill et al., 2023), and show similar dynamics in sparse parity tasks (Merrill et al., 2023). The addition of regularization has been shown to strongly affect grokking in certain scenarios (Power et al., 2022; Liu et al., 2023). This connection may be attributed to weight decay (WD), for instance, improving generalization (Krogh and Hertz, 1991), though this property is not yet fully understood (Zhang et al., 2018). We incorporate WD in our setup and study its effects on grokking analytically, showing that it can either suppress or enhance grokking, depending on the number of network parameters and number of training samples.

Key related works, most closely tied with our own, are Liu et al. (2022; 2023) and Žunkovič and Ilievski (2022); Gromov (2023). Liu et al. (2022) show perfect generalization on a non-modular addition task when enough data determines the structured representation. Liu et al. (2023) relate grokking to memorization dynamics. Žunkovič and Ilievski (2022); Gromov (2023) analyze solvable models displaying grokking and relate results to latent-space structure formation. Our work employs a similar setup but derives grokking dynamics from a random matrix theory perspective relating dataset properties to the empirical covariance matrix.

**Linear Estimators in High Dimensions**  A growing body of work has focused on deriving exact solutions for linear estimators trained on Gaussian data, particularly in the context of random feature models. The dynamics are often described in the gradient flow limit, which we employ in this work. Building on statistical physics methods, Sompolinsky et al. (1988); Advani and Saxe (2017) provided an analytical characterization of the dynamics of learning in linear neural networks under gradient descent, both in shallow and in deep networks Saxe et al. (2014). Their mean-field analysis precisely tracks the evolution of the training and generalization errors, similar to Richards et al. (2021); Mignacco et al. (2021); Mignacco and Urbani (2022); Paquette et al. (2022). More recently, Bodin and Macris (2022) further studied the dynamics of generalization under gradient descent for the Gaussian covariate model, corroborating the presence of epoch-wise descent structures. In the context of least squares estimation and multiple layers, Loureiro et al. (2022); Goldt et al. (2020) analyzed the gradient flow dynamics and long-time behavior of the training and generalization errors.

The tools from random matrix theory and statistical mechanics employed in these analyses allow precise tracking of the generalization curve and transitions thereof, akin to Dobriban and Wager (2015). Our work adopts a similar theoretical framing to study the interplay between model capacity, overparameterization, and gradient flow optimization in determining generalization performance.

## 3 TRAINING DYNAMICS IN A LINEAR TEACHER-STUDENT SETUP

The majority of our results are derived for a simple student-teacher model Seung et al. (1992), where the inputs are identical independently distributed (iid) normal variables. We draw $N_{\text{tr}}$ training samples from a standard Gaussian distribution $x_i \sim \mathcal{N}(0, \boldsymbol{I}_{d_{\text{in}} \times d_{\text{in}}})$, and the teacher model generates output labels. The student is trained to mimic the predictions of the teacher, which we take as perfect.

The teacher and student models, which we denote by $T$ and $S$ respectively, share the same architecture. As we show below, Grokking can occur even for the simplest possible network function, which is a linear Perceptron with no biases, or in other words – a simple linear transformation. The loss function is the standard MSE loss. Our analyses are done in the regime of large input dimension and large sample size, i.e., $d_{\text{in}}, N_{\text{tr}} \to \infty$, where the ratio $\lambda \equiv d_{\text{in}}/N_{\text{tr}} \in \mathbb{R}^+$ kept constant.

Following the construction presented in Liu et al. (2023), we can convert this regression problem into a classification task by setting a threshold $\epsilon > 0$ and defining a sample to be correctly classified if the prediction error is less than $\epsilon$. The student model is trained with the full batch Gradient Descent (GD) optimizer for $t$ steps with a learning rate $\eta$, which may also include a weight decay parameter $\gamma$. The training loss function is given by

$$\mathcal{L}_{\text{tr}} = \frac{1}{N_{\text{tr}} d_{\text{out}}} \sum_{i=1}^{N_{\text{tr}}} \|(S - T)^T x_i\|^2 = \frac{1}{d_{\text{out}}} \text{Tr}\left[D^T \Sigma_{\text{tr}} D\right], \qquad D \equiv S - T. \qquad (1)$$

where $S, T \in \mathbb{R}^{d_{\text{in}} \times d_{\text{out}}}$ are the student and teacher weight matrices, $\Sigma_{\text{tr}} \equiv \frac{1}{N_{\text{tr}}} \sum_{i=1}^{N_{\text{tr}}} x_i x_i^T$ is the $d_{\text{in}} \times d_{\text{in}}$ empirical data covariance, or Gram matrix for the *training* set, and we define $D$ as the difference between the student and teacher matrices. The elements of $T$ and $S$ are drawn at initialization from a normal distribution $S_0, T \sim \mathcal{N}(0, 1/(2d_{\text{in}} d_{\text{out}}))$. We do not include biases in the student or teacher weight matrices, as they do not affect centrally distributed data.

Similarly, the generalization loss function is defined as its expectation value over the input distribution, which can be approximated by the empirical average over $N_{\text{gen}}$ randomly sampled points

$$\mathcal{L}_{\text{gen}} = \mathbb{E}_{x \sim \mathcal{N}}\left[\frac{1}{d_{\text{out}}} \|(S - T)^T x\|^2\right] = \frac{1}{d_{\text{out}}} \text{Tr}\left[D^T \Sigma_{\text{gen}} D\right] = \frac{1}{d_{\text{out}}} \|D\|^2. \qquad (2)$$

Here $\Sigma_{\text{gen}}$ is the covariance of the generalization distribution, which is the identity. Note that in practice the generalization loss is computed by a sample average over an independent set, which is not equal to the analytical expectation value. The gradient descent equations at training step $t$ are

$$\nabla_D \mathcal{L}_{\text{tr}} = \frac{2}{d_{\text{out}}} \Sigma_{\text{tr}} D, \qquad D_{t+1} = \left(\boldsymbol{I} - \frac{2\eta}{d_{\text{out}}} \Sigma_{\text{tr}}\right) D_t - \frac{\eta\gamma}{d_{\text{out}}}\left(D_t + T\right), \qquad (3)$$

where $\gamma \in \mathbb{R}^+$ is the weight decay parameter, and $\boldsymbol{I} \in \mathbb{R}^{d_{\text{in}} \times d_{\text{in}}}$ is the identity.

It is worthwhile to emphasize the difference between Eq. (1) and Eq. (2), since the distinction between sample average and analytical expectation value is crucial to our analyses. In training, Eq. (1), we compute the loss over a fixed dataset whose covariance, $\Sigma_{\text{tr}}$, is non-trivial. The generalization loss is defined as the expectation value over the input distribution, which has a trivial covariance by assumption, $\Sigma_{\text{gen}} = \boldsymbol{I}$. Even if it is computed in practice by averaging over a finite sample with a non-trivial covariance, it is independent of the training dynamics and the sample average will converge to the analytical expectation with the usual $\sqrt{N}$ scaling. This is *not true* for the training loss, since the training dynamics will guide the network in a direction that minimizes the empirical loss with respect to the fixed covariance $\Sigma_{\text{tr}}$. This assertion is numerically verified below, as we compare the generalization loss, practically computed by sample averaging, to the analytical result of Eq. (2).

### 3.1 WARMUP: THE SIMPLEST MODEL

#### 3.1.1 TRAIN AND GENERALIZATION LOSS

Before analyzing the dynamics of the general linear model, we start with a simpler setting that captures the most important aspects of the full solution. Concretely, here we set $d_{\text{out}} = 1$, reducing $S, T \in \mathbb{R}^{d_{\text{in}}}$ from matrices to vectors, and assume no weight decay $\gamma = 0$.

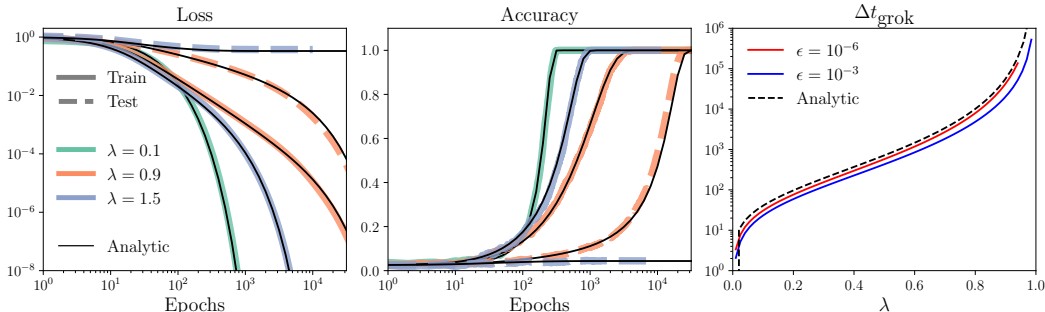

Figure 1: Grokking as a function of $\lambda$. **Left:** Empirical results for training (dashed) and test (solid) losses, for $\lambda = 0.1, 0.9, 1.5$ against analytical solutions (black). For $\lambda = 0.1$ the test and train curves are indistinguishable. **Center:** Similar comparison for the accuracy functions. **Right:** Grokking time as a function of $\lambda$, for different values of the threshold parameter $\epsilon$ (red, blue), and shown the analytic solution Eq. (11) (dashed black). Here we use GD with $\eta = \eta_0 = 0.01, d_{\rm in} = 10^3, d_{\rm out} = 1, \epsilon = 10^{-3}$.

Eq. (3) can be solved in the gradient flow limit of continuous time, setting $\eta = \eta_0 dt$ and $dt \to 0$, resulting in

$$\dot{D}(t) = -2\eta_0 \Sigma_{\rm tr} D(t) \quad \to \quad D(t) = e^{-2\eta_0 \Sigma_{\rm tr} t} D_0, \tag{4}$$

where $D_0$ is simply the difference between teacher and student vectors at initialization. It follows that the empirical losses, calculated over a dataset admit closed-form expressions as

$$\mathcal{L}_{\rm tr} = D_0^T e^{-4\eta_0 \Sigma_{\rm tr} t} \Sigma_{\rm tr} D_0, \qquad \mathcal{L}_{\rm gen} = D_0^T e^{-4\eta_0 \Sigma_{\rm tr} t} D_0. \tag{5}$$

These expressions for the losses are exact. To proceed, we need to know the Gram matrix of the training dataset, which is the empirical covariance of a random sample of Gaussian variables. It is known that eigenvectors of $\Sigma_{\rm tr}$ are uniformly distributed on the unit sphere in $\mathbb{R}^{d_{\rm out}}$ and its eigenvalues, $\nu_i$, follow the Marchenko-Pastur (MP) distribution (Marčenko and Pastur, 1967),

$$p_{\rm MP}(\nu)d\nu = \left(1 - \frac{1}{\lambda}\right)^+ \delta_0 + \frac{\sqrt{(\lambda_+ - \nu)(\nu - \lambda_-)}}{2\pi\lambda\nu} I_{\nu \in [\lambda_-, \lambda_+]} d\nu, \tag{6}$$

where $\delta_\nu$ is the Dirac mass at $\nu \in \mathbb{R}$, we define $x^+ = \max\{x, 0\}$ for $x \in \mathbb{R}$, and $\lambda_\pm = (1 \pm \sqrt{\lambda})^2$.

Since the directions of both $D$ and the eigenvectors of $\Sigma_{\rm tr}$ are uniformly distributed, we make the approximation that the projection of $D$ on all eigenvectors is the same, which transforms Eq. (5) to the simple form

$$\mathcal{L}_{\rm tr} \approx \frac{1}{d_{\rm in}} \sum_i e^{-4\eta_0 \nu_i t} \nu_i, \qquad\qquad \mathcal{L}_{\rm gen} \approx \frac{1}{d_{\rm in}} \sum_i e^{-4\eta_0 \nu_i t}. \tag{7}$$

It is seen that these sums are the empirical average over the function $e^{-4\eta_0 \nu t} \nu$, if $\nu$ follows the MP distribution. This can be well approximated by their respective expectation values,

$$\mathcal{L}_{\rm tr}(\eta_0, \lambda, t) \approx \mathbb{E}_{\nu \sim \rm MP(\lambda)}\left[\nu e^{-4\eta_0 \nu t}\right], \qquad \mathcal{L}_{\rm gen}(\eta_0, \lambda, t) \approx \mathbb{E}_{\nu \sim \rm MP(\lambda)}\left[e^{-4\eta_0 \nu t}\right]. \tag{8}$$

The evolution of these loss functions is dictated by the MP distribution, which exhibits distinct behaviors for $\lambda < 1$ and $\lambda > 1$. For $\lambda < 1$, the first term in Eq. (6) vanishes, the distribution has no null eigenvalues and so $\mathcal{L}_{\rm tr}, \mathcal{L}_{\rm gen}$ both are driven to 0 at $t \to \infty$, implying that perfect generalization is always obtained eventually. On the other hand, for $\lambda > 1$, Eq. (6) develops several zero eigenvalues, corresponding to flat directions in the training Gram matrix. In this case, while $\mathcal{L}_{\rm tr}$ is driven to 0, since $\nu e^{-4\eta_0 \nu t}|_{\nu=0} = 0$, the generalization loss $\mathcal{L}_{\rm gen}$ does not vanish, as $e^{-4\eta_0 \nu t}|_{\nu=0} = 1$ contributes a nonzero constant $1 - 1/\lambda$ to the loss, preventing perfect generalization. When $d_{\rm out} = 1$, the two regimes correspond to underparameterization ($\lambda < 1$) and overparameterization ($\lambda > 1$).

In Fig. 1 we show that these analytical predictions are in excellent agreement with numerical experiments, with no fitting parameters, in both regimes.

We also note that the expectation value of $\mathcal{L}_{\rm tr}$ in Eq. (8) admits a closed form solution, $\mathcal{L}_{\rm tr} = e^{-4\eta_0(\lambda+1)t} {}_0\tilde{F}_1\left(2; 16\eta_0^2 t^2 \lambda\right)$, where ${}_0\tilde{F}_1\left(a; z\right) = {}_0F_1(a; z)\Gamma(a)$ is the regularized confluent hypergeometric function. We could not find a closed form expression for $\mathcal{L}_{\rm gen}$, but approximate expressions for the expectation value can be derived for the late time behavior, cf. Appendix B.

### 3.1.2 TRAIN AND GENERALIZATION ACCURACY

Next, we describe the evolution of the training and generalization accuracy functions. As described above, in the construction of Liu et al. (2023) the accuracy $\mathcal{A}$ is defined as the (empirical) fraction of points whose prediction error is smaller than $\epsilon$, $\mathcal{A} = \frac{1}{N} \sum_{i=1}^{N} \Theta(\epsilon - (D^T(t)x_i)^2)$, where $\Theta$ is the Heaviside step function. We define $z = D^T x \in \mathbb{R}$, which is normally distributed with standard deviation $D^T \Sigma D = \mathcal{L}$, where $\Sigma$ is the covariance of $x$ (that is, $\Sigma_{\text{tr}}$ for training and $I$ for generalization). Then, in the limit of large sample sizes, the empirical averages converge to

$$\mathcal{A} \xrightarrow[N \to \infty]{} 2 \Pr\left(|z| \leq \sqrt{\epsilon}\right) = \text{Erf}\left(\sqrt{\frac{\epsilon}{2\mathcal{L}}}\right) , \tag{9}$$

where $\mathcal{A}, \mathcal{L}$ stand for train and generalization measures. This result implies that the increase in accuracy in the late stages of training can be simply mapped to the decrease of the loss below $\epsilon$. Writing the accuracy as an explicit function of the loss allows an exact calculation of the grokking time, and of whether grokking occurs at all.

### 3.1.3 GROKKING TIME

In this framework, grokking is simply the phenomenon in which $\mathcal{L}_{\text{tr}}$ drops below $\epsilon$ before $\mathcal{L}_{\text{gen}}$ does. To understand exactly when these events happen, in Appendix B we derive approximate results in the long time limit, $\eta_0 t \gg \sqrt{\lambda}$, showing that

$$\mathcal{L}_{\text{tr}} \simeq \frac{\exp\left[-4\eta_0\left(1 - \sqrt{\lambda}\right)^2 t\right]}{16\sqrt{\pi}\lambda^{3/4}(\eta_0 t)^{3/2}} , \qquad \mathcal{L}_{\text{gen}} \simeq \mathcal{L}_{\text{tr}} \times \left(1 - \sqrt{\lambda}\right)^{-2} . \tag{10}$$

We define grokking time as the time difference between the training and generalization accuracies reaching $\text{Erf}(\sqrt{2}) \approx 95\%$, obtained when each loss satisfies $\mathcal{L}(t^*) = \epsilon/4$. In terms of the loss functions, we show in Appendix B that solving for the difference between $t^*_{\text{gen}} - t^*_{\text{tr}}$, and expanding the result in the limit of $\epsilon \ll 1$, one obtains an analytic expression for the grokking time difference

$$\Delta t_{\text{grok}} = t^*_{\text{gen}} - t^*_{\text{tr}} \simeq -\frac{\log\left(1 - \sqrt{\lambda}\right)}{2\eta_0\left(1 - \sqrt{\lambda}\right)^2}. \tag{11}$$

Eq. (11) indicates that the maximal grokking time difference occurs near $\lambda \simeq 1$, where the grokking time diverges quadratically as $\Delta t_{\text{grok}}(\lambda \to 1) \sim \frac{1}{\eta_0(\lambda-1)^2} \log\left(\frac{4}{(1-\lambda)^2}\right)$. On the other hand, it vanishes for $\lambda \simeq 0$, which means $N_{\text{tr}} \gg d_{\text{in}}$ and $\Sigma_{\text{tr}}$ approaches the identity, as expected. These predictions are verified in Fig. 1(right).

**Effects of Initialization and Label Noise:** We briefly comment on the effect of choosing a different initialization for the student weights compared to the teacher weights, which is discussed in Liu et al. (2023), as well as adding training label noise. In the first setup, rescaling the student weights $S \to \alpha S$ leads to a trivial rescaling of both the training and generalization loss functions as $\mathcal{L} \to \frac{1+\alpha^2}{2}\mathcal{L}$, which is tantamount to choosing a different threshold parameter $\epsilon \to \frac{2\epsilon}{1+\alpha^2}$, leaving the results unchanged. In the case of training label noise $y \to y + \delta$ , where $\delta \sim \mathcal{N}(0, \sigma_\delta^2)$, the student dynamics don't change, but the training loss function would receive a constant contribution, proportional to the noise variance $\sigma_\delta^2$, as detailed in Appendix C. It is then possible to tune $\sigma_\delta^2$ and the threshold parameter $\epsilon$, such that the noise can induce grokking precisely when test accuracy saturates. This effect could be wrongly interpreted as "self-correction" (Liu et al., 2022).

### 3.2 INTERPRETATION AND INTUITION

We conclude this section by summarizing and interpreting the analytical results for the simple 1-layer linear network with a scalar output and MSE loss. In this setting, the loss, which is an empirical average over a finite sample, is given by the norm of $D = S - T$, as measured by the metric defined by the covariance of the sample, $\mathcal{L} = D^T \Sigma D$. While the generalization covariance is the identity by

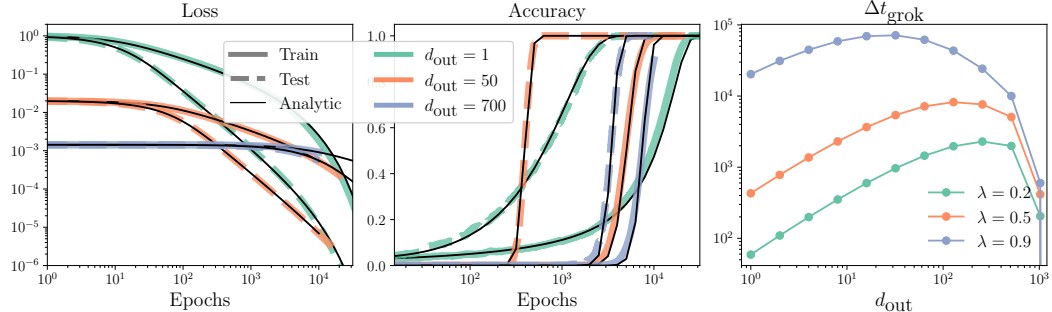

Figure 2: Effects of the output dimension $d_{\text{out}} > 1$ on grokking. **Left:** Empirical results for training (dashed) and generalization (solid) losses, for $d_{\text{out}} = 1, 50, 700$ (blue, red, violet) against analytical solutions (black), for $\lambda = 0.9$. **Center:** Similar comparison for the accuracy functions. **Right:** The grokking time as a function of $d_{\text{out}}$, for different values of $\lambda$. Different solid curves are numerical solutions for the expressions given in Section 3.3.1. Training is done using GD with $\eta = \eta_0 = 0.01, d_{\text{in}} = 10^3, \epsilon = 10^{-3}$.

construction, the train covariance only approaches the identity in the limit $N_{\text{tr}} \gg d_{\text{in}}$, and otherwise follows the Marchenko-Pastur distribution.

The training gradients point to a direction that minimizes the training loss, which is $\|D\|_{\Sigma_{\text{tr}}}$, and in the long time limit, it vanishes exponentially. This must imply that the generalization loss, $\|D\|_{I}$, which is the norm of the same vector but calculated with respect to a different metric, also vanishes exponentially but somewhat slower. Since in this setting the accuracy is a function of the loss, grokking is identified as the difference between the times that the training and generalization losses fall below the fixed threshold $\epsilon/4$. We note that the fact that the accuracy is an explicit function of the loss is a useful peculiarity of this model. In more general settings it is not the case, though it is generally expected that low loss would imply high accuracy.

However, it is noteworthy that nothing particularly interesting is happening at this threshold, and the loss dynamics are oblivious to its existence. In other words, grokking in this setting, as reported previously by Liu et al. (2023), is an artifact of the definition of accuracy and does not represent a transition from "memorization" to "understanding", or any other qualitative increase in any generalization abilities of the network.

Our analysis can be easily extended to include other effects in more complicated scenarios, which we detail below. In all these generalizations the qualitative interpretation remains valid.

## 3.3 VARIANTS

### 3.3.1 THE EFFECT OF $d_{\text{out}}$

We first extend our analysis to the case $d_{\text{out}} > 1$. The algebra in this case is similar to what was shown in Section 3.1. We provide the full derivation in Appendix D and report the main results here. The loss evolution follows the same functional form as Eq. (8), with the replacement $\eta_0 \to \eta_0/d_{\text{out}}$, and a correction to Eq. (9):

$$\mathcal{L}^{d_{\text{out}} \neq 1}(\eta_0, \lambda, t) = \frac{1}{d_{\text{out}}} \mathcal{L}^{d_{\text{out}} = 1}\left(\frac{\eta_0}{d_{\text{out}}}, \lambda, t\right), \qquad \mathcal{A} = 1 - \frac{\Gamma\left(\frac{d_{\text{out}}}{2}, \frac{d_{\text{out}}}{2\mathcal{L}}\epsilon\right)}{\Gamma\left(\frac{d_{\text{out}}}{2}\right)}, \qquad (12)$$

Here, $\Gamma(a, z) = \int_a^\infty e^{-t} t^{z-1} dt$ is the incomplete gamma function, and $\Gamma(z) = \int_0^\infty e^{-t} t^{z-1} dt$ is the gamma function. These relation hold for $\mathcal{A}_{\text{gen/tr}}$ and $\mathcal{L}_{\text{gen/tr}}$ separately.

We note that the expression for $\mathcal{A}$ stems from the fact that $\|z\|^2 = \|D^T x\|^2$ now follows a $\chi^2$ distribution and not a normal distribution. It is seen that $\mathcal{A}$ is still an explicit function of $\mathcal{L}$, albeit somewhat more complicated.

The effects of $d_{\text{out}} > 1$ can be read from Eq. (12), and are twofold. Firstly, the accuracy rapidly approaches 1 as the output dimension $d_{\text{out}}$ increases, for any value of $\mathcal{L}$ and $\epsilon$. This implies that in the limit of $d_{\text{out}} \to \infty$, both training and generalization accuracies must be close to 100% shortly after initialization and no grokking occurs. Secondly, the learning rate $\eta_0$ becomes effectively smaller as

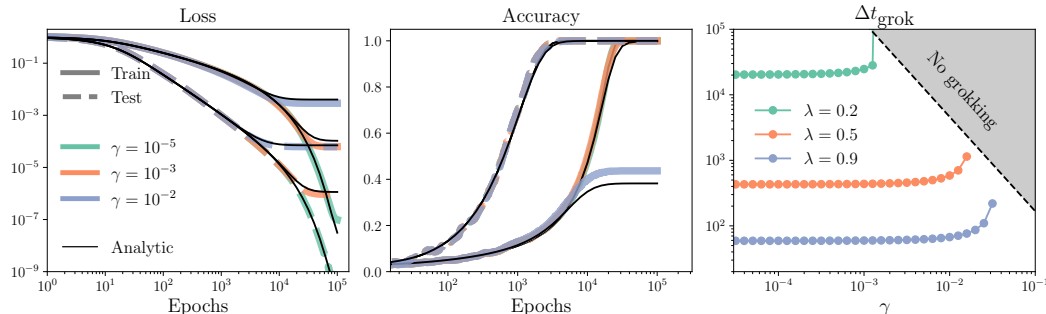

Figure 3: Effects of weight decay ($\gamma$) on grokking. **Left:** Empirical results for training (dashed) and generalization (solid) losses, for $\gamma = 10^{-5}, 10^{-3}, 10^{-2}$ (blue, red, violet) against analytical solutions (black), for $\lambda = 0.9$. **Center:** Similar comparison for the accuracy functions. **Right:** The grokking time as a function of $\gamma$, for different values of $\lambda$. Different solid curves are numerical solutions for the expressions given in Section 3.3.1, while the shaded gray region corresponds to training/generalization saturation, without perfect generalization. Training is done using GD with $\eta = \eta_0 = 0.01, d_{\text{in}} = 10^3, d_{\text{out}} = 1, \epsilon = 10^{-3}$

$d_{\text{out}}$ grows, implying that the overall time scale of convergence for both training and generalization accuracies increases, leading to a higher grokking time. These two competing effects, along with the monotonicity of the loss functions, give rise to a non-monotonic dependence of the grokking time on $d_{\text{out}}$, which attains a maximum at a specific value $d_{\text{out}}^{\text{max}}$, as can be seen in Fig. 2.

### 3.3.2 THE EFFECT OF WEIGHT DECAY

We consider first the case of nonzero WD in the simpler case of $d_{\text{out}} = 1$. Incorporating weight decay amounts to adding a regularization term at each gradient descent timestep, modifying Eq. (3) to

$$D_{t+1} = D_t - 2\eta \left( \Sigma_{\text{tr}} + \frac{\gamma}{2}I \right) D_t - \eta\gamma T, \tag{13}$$

where $\gamma \in \mathbb{R}^+$ is the weight decay parameter. The calculations are straightforward and detailed in Appendix E, the result being that Eq. (8) should be modified to read

$$\mathcal{L} = \frac{1}{2}\mathbb{E}_{\nu \sim \text{MP}(\lambda)} \left[ \left( e^{-4\eta_0 \left( \nu + \frac{1}{2}\gamma \right)t} + \left( \frac{e^{-2\eta_0 \left( \nu + \frac{1}{2}\gamma \right)t}\nu + \frac{1}{2}\gamma}{\nu + \frac{1}{2}\gamma} \right)^2 \right) q \right], \tag{14}$$

where $q = \nu$ for the training loss $q = 1$ for generalization. Compared to Eq. (8), it is seen that the main effect of WD is a shift in the effective spectrum $\nu \to \nu + \frac{1}{2}\gamma$, as can be expected from the second term in Eq. (13). Since $\gamma$ only affects the gradient but not the accuracy, the expression in Eq. (9) of $\mathcal{A}$ as a function of $\mathcal{L}$, remains unchanged.

It is instructive to analyze Eq. (14) separately for the under and overparameterized regimes. When $\lambda < 1$, the MP distribution has no null eigenvalues, and the losses begin by decaying exponentially. We can study the grokking behavior by examining the late time limit, i.e. $t \to \infty$, in which the exponential terms decay, and approximating for small $\gamma \ll 1$, we obtain the asymptotic expressions

$$\mathcal{L}_{\text{tr}} \simeq \frac{\gamma^2}{4(1-\lambda)}, \qquad \mathcal{L}_{\text{gen}} \simeq \frac{\gamma^2}{4(1-\lambda)^3}, \qquad \Delta t_{\text{grok}} \simeq \frac{\log\left(1+\sqrt{\lambda}\right)}{2\eta_0\left(1-\sqrt{\lambda}\right)^2}. \tag{15}$$

This result means that the generalization loss has a higher asymptotic value than the training loss. Thus, there is a value of $\epsilon$ below which perfect generalization cannot be obtained. For $\epsilon$ above this threshold, WD has no effect, and below it, the grokking time decreases as given by Eq. (15).

In the overparameterized regime, where $\lambda > 1$, the MP distribution necessarily contains vanishing eigenvalues, which, as shown in Fig. 1, cause the generalization loss to plateau. Introducing weight decay changes this picture somewhat, causing the null eigenvalues to be shifted by a factor of $\gamma/2$ and ensuring that better generalization performance is reached. Still, the late time behavior is the same as Eq. (15), following the same arguments as discussed above. We note that in this case, the relevant timescale of the generalization loss is determined by $1/\gamma$, leading to suppressing grokking, as observed by Liu et al. (2023), though their regime lies in the gray region of Fig. 3.

The grokking time behaviors for various values of $\gamma$ are clearly demonstrated in Fig. 3. In addition, Fig. 7 in the appendix shows the combined effects of $\lambda$, $\gamma$ and $d_{\text{out}}$.

# 4 GENERALIZATIONS

## 4.1 2-LAYER NETWORKS

Our analysis can be generalized to multi-layer models. Here, we consider the addition of a single hidden layer, where the teacher network function is $f(x) = T_1^T \sigma \left( T_0^T x \right)$, where $T_0 \in \mathbb{R}^{d_{in} \times d_h}$, $T_1 \in \mathbb{R}^{d_h \times d_{out}}$, $\sigma$ is an entry-wise activation function and $d_h$ is the width of the hidden layer. Similarly, the student network is defined by two matrices $S_0(t), S_1(t)$. The empirical training loss reads

$$\mathcal{L}_{tr} = \frac{1}{N_{tr} d_{out}} \sum_{i=1}^{N_{tr}} \left[ S_1^T \sigma \left( (S_0)^T x_i \right) - T_1^T \sigma \left( T_0^T x_i \right) \right]^2 . \tag{16}$$

In this setup the weights are drawn at initialization from normal distributions $S_0(t = 0), T_0 \sim \mathcal{N}(0, 1/(2d_{in} d_h))$ and $S_1(t = 0), T_1 \sim \mathcal{N}(0, 1/(2d_{out} d_h))$.

As a solvable model, we consider first the case of linear activation, $\sigma(z) = z$, i.e., a two-layer linear network. In this case, we can define $T = T_0 T_1 \in \mathbb{R}^{d_{in} \times d_{out}}$ as we did in the previous sections, since the teacher weights are not updated dynamically. Similar to Eqs. (1) and (2), under the definition $D(t) = S_0(t) S_1(t) - T$, we show in Appendix F that the gradient flow equations for the system are

$$\partial_t D = -\frac{2\eta_0}{d_{out}^2} h(t) \Sigma_{tr} D(t) , \qquad \partial_t h = -8\eta_0 (T + D(t))^T \Sigma_{tr} D(t). \tag{17}$$

Here, $h(t) = \frac{1}{2} \left( \|S_0(t)\|^2 + \|S_t(t)\|^2 \right)$. Although Eq. (17) describes a set of coupled equations, we note that the solution for $h(t)$ can be simplified when considering the limit of small $\eta_0 \ll 1$, as we may ignore the time evolution and consider the trace (or kernel) as fixed to its initialization value, which is $h \simeq 1/2$ for $d_h \gg d_{out}$. In this case, the solutions for $\mathcal{L}$ are a simple modification of the one given in the previous sections, with the replacement $\eta_0 \to \eta_0/(2d_{out}^2)$. Subsequently, the training/generalization performance metrics are

$$\mathcal{L}^{2-layer} (\eta_0, \lambda, t) = \|D_0\|^2 \mathcal{L}^{1-layer} \left( \frac{\eta_0}{2d_{out}^2}, \lambda, t \right) , \tag{18}$$

and $\mathcal{A}_{tr/gen}$ is a function of $\mathcal{L}_{tr/gen}$, respectively, as given in Eq. (12). We note that this setup is generically overparameterized for large $d_h$, regardless of $d_{out}$ and for any $\lambda$. In this sense, all of the results previously derived for $\lambda < 1$ should hold, and grokking occurs as discussed in the previous sections. We experimentally verify that Eq. (18) correctly predicts the performance metrics and their dynamics in Fig. 4 (left column).

## 4.2 NON-LINEAR ACTIVATIONS

The final extension of our work is to consider the network in Section 4.1, but choose nonlinear activation functions for the hidden layer. In the limit of large $d_h \gg 1$, we expect the network to begin to linearize, eventually converging to the Neural Tangent Kernel (NTK) regime (Lee et al., 2019). In this regime, the results in Section 4.1 should hold, up to a redefinition of the kernel which depends on the nonlinearity.

In Fig. 4 (right column), we show that the dynamics of a 2-layer MLP (1000-200-5) with $\tanh$ activations is well approximated by Eqs. (17) and (18), empirically verifying that our predictions hold beyond the linear regimes, in some cases.

# 5 DISCUSSION

We have shown that grokking, at least in the sense of late generalization, can occur in simple linear teacher-student settings and provided explicit analytical solutions for the training and generalization loss and accuracy dynamics during training. The predictions, which strictly apply in the gradient-flow limit and for large sample sizes, were corroborated against numerical experiments and provided an excellent description of the dynamics

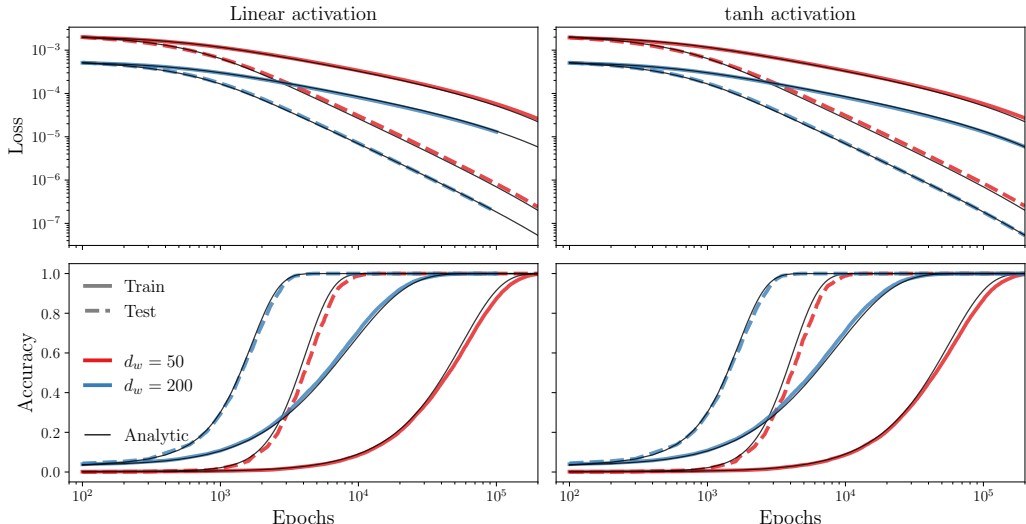

Figure 4: 2-Layer network and nonlinearities. **Left:** Empirical results for training (dashed) and generalization (solid) losses/accuracies (*top/bottom*), for a two-layer MLP (1000-$d_h$-5) with linear activations and $d_h = 50, 200$ (blue, red), against analytical solutions (black). **Right:** Similar results for an identical network with $\tanh$ activations. In both cases, training is done using full batch gradient descent with $\eta = \eta_0 = 0.01, d_{in} = 1000, d_{out} = 5, \epsilon = 10^{-4}$.

A main point of this work is that grokking in these cases is an artifact of the accuracy metric and does not represent "understanding" in any meaningful way. Moreover, this scenario does not capture some of the observed phenomenology of grokking, which sometimes features non-monotonic loss evolution and special weight structures as observed in Gromov (2023); Liu et al. (2022), both of which are not present in our setup. Therefore, the proposed mechanism is clearly not a general theory of grokking: the closed-form solutions for the dynamics clearly pertain only to this specific setup, and depend on the spectral properties of the covariance. We thus do not expect that they will generalize "as is" in other contexts.

However, parts of the proposed mechanism may be useful in understanding certain aspects of grokking even in realistic scenarios. The core property of the proposed underlying mechanism of grokking is the fact that under GD dynamics, the accuracy is simply a function of the loss (cf. Eq. (9) and its generalization Eq. (12)), and importantly – a rapidly changing function, with a threshold that depends on the data covariance (in our case, this is the condition $x^T \Sigma x < \epsilon$). In this sense, late generalization is an artifact of the metric and does not represent any dynamical property or special weight structure. We suspect that this core property, as well as some properties of the analytical predictions, may be observed in more realistic scenarios. In Appendix G we present preliminary results regarding this conjecture, in the context of the classic modular addition task discussed in Gromov (2023). It is shown that the proposed picture is consistent with the numerical experiments.

The application of the proposed mechanism to modeling grokking in realistic settings will be studied in future research. In addition, it would be interesting to study how universal are our results when taking into account finite learning rates, realistic data covariance structures, different choices of optimizers or losses, and other effects.

# 6 ACKNOWLEDGEMENTS

We thank Boaz Barak, Nadav Cohen and Andrey Gromov for fruitful discussions. YBS was supported by research grant ISF 1907/22 and Google Gift grant. NL would like to thank the Milner Foundation for the award of a Milner Fellowship. This work was initiated in part at Aspen Center for Physics, which is supported by National Science Foundation grant PHY-2210452.

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

## A EXPERIMENTAL DETAILS

In all of our experiments, we employ a teacher-student model with shared architecture for both teacher and student. The training data consists of a fixed number of training samples quoted in the main text for each experiment, drawn from a normal distribution $\mathcal{N}(0, \boldsymbol{I})$. All experiments are done on MLPs using MSE loss with the default definitions employed by PyTorch. The exact details of each MLP depend on the setup and are quoted in the main text. We train with full batch gradient descent, in all instances. We depart from the default weight initialization of PyTorch, using $w \sim \mathcal{N}(0, 1/(2d_{l-1}d_l)$ for each layer, where $d_{l-1}$ is in the input dimension coming from the previous layer and $d_l$ is the output dimension of the current layer.

## B DERIVATION OF THE GROKKING TIME DIFFERENCE

Here, we provide the full derivation for the grokking time difference presented in Eq. (11). Our starting point is the exact solution for the training loss in $d_{\text{out}} = 1$ case for $\lambda < 1$, given by

$$\mathcal{L}_{\text{tr}} = e^{-4\eta_0(\lambda+1)t} \, {}_0\tilde{F}_1\left(2; 16\eta_0^2 t^2 \lambda\right), \tag{19}$$

where ${}_0\tilde{F}_1\left(a; z\right) = {}_0F_1(a; z)\Gamma(a)$ is the regularized confluent hypergeometric function. We also note the relation

$$\frac{d\mathcal{L}_{\text{gen}}}{dt} = -4\eta_0 \mathcal{L}_{\text{tr}}, \tag{20}$$

which we will use to relate training and generalization loss functions. Since we are interested in the late time behavior, where grokking occurs, we expand the training loss for $\eta_0 t \gg \sqrt{\lambda}$, which is given at leading order by

$$\mathcal{L}_{\text{tr}} \simeq \frac{e^{-4\eta_0\left(1-\sqrt{\lambda}\right)^2 t}}{16\sqrt{\pi}\lambda^{3/4}(\eta_0 t)^{3/2}}. \tag{21}$$

Plugging in the result of Eq. (21) into Eq. (20) and integrating over time, we find the expression for the generalization loss at late times is given by

$$\mathcal{L}_{\text{gen}} \simeq \frac{\sqrt{\eta_0 t}e^{-4\eta_0\left(1-\sqrt{\lambda}\right)^2 t}}{2\sqrt{\pi}\eta_0 t\lambda^{3/4}} - \frac{\left(1-\sqrt{\lambda}\right)\Gamma\left(\frac{1}{2}, 4\eta_0 t\left(1-\sqrt{\lambda}\right)^2\right)}{\sqrt{\pi}\lambda^{3/4}}, \tag{22}$$

where $\Gamma(a, z) = \int_z^\infty dt e^{-t}t^{a-1}$ is the incomplete gamma function. Expanding the result further for late times, we arrive at the result quoted in Eq. (10). In Fig. 5, we show the approximate late-time solutions against the exact solutions. The approximations hold quite well even at somewhat early times, and become increasingly more accurate for later epochs.

With the loss functions at hand, we turn to the grokking time itself. As described in the main text, we define the grokking time as the time difference between the training and generalization accuracies reaching $\text{Erf}(\sqrt{2}) \approx 95\%$, obtained when each loss satisfies $\mathcal{L}(t^*) = \epsilon/4$. Solving this equation for each loss separately, in the late time limit, gives the following expressions for the training and generalization times

$$t_{\text{tr}}^* \simeq \frac{3}{8\eta_0\left(1-\sqrt{\lambda}\right)^2}\mathcal{W}\left(\frac{2\,2^{2/3}\sqrt[3]{\frac{\lambda^{3/2}}{\pi} + \frac{1}{\pi\lambda^{3/2}} - \frac{6\lambda}{\pi} + \frac{15\sqrt{\lambda}}{\pi} - \frac{6}{\pi\lambda} + \frac{15}{\pi\sqrt{\lambda}} - \frac{20}{\pi}}}{3\epsilon^{2/3}}\right), \tag{23}$$

$$t_{\text{gen}}^* \simeq \frac{3}{8\eta\left(1-\sqrt{\lambda}\right)^2}\mathcal{W}\left(\frac{2^{5/3}\sqrt[3]{\frac{1}{\pi\lambda^{3/2}} + \frac{1}{\pi\sqrt{\lambda}} - \frac{2}{\pi\lambda}}}{3\epsilon^{2/3}}\right), \tag{24}$$

where $\mathcal{W}(z)$ is the Lambert W function, which solves the equation $\mathcal{W}e^{\mathcal{W}} = z$, also known as the product-log function. As the argument of both training and generalization times are large, we

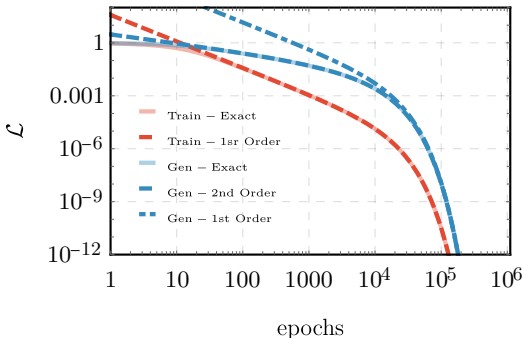

Figure 5: Exact training and generalization losses against approximate solutions at late times. In **pink/light blue**, we show the solutions of Eq. (8). In **dashed red** is Eq. (21), in **dashed blue**, we show Eq. (22), while **dotted-dashed blue** is the solution given in the main text, Eq. (10). Clearly, the asymptotic behavior matches the exact solutions. Here, $\eta_0 = 0.01, \lambda = 0.9, d_{\text{out}} = 1$.

can expand the Lambert function to leading order in $z$ as $\mathcal{W}(z) \simeq \log(z)$. Taking the difference $\Delta t_{\text{grok}} = t^*_{\text{gen}} - t^*_{\text{tr}}$ and expanding to leading order in $\epsilon \ll 1$, we obtain the final expression

$$\Delta t_{\text{grok}} = t^*_{\text{gen}} - t^*_{\text{tr}} \simeq \frac{\log\left(\frac{1}{1-\sqrt{\lambda}}\right)}{2\eta_0 \left(1 - \sqrt{\lambda}\right)^2} + \frac{3}{8\eta \left(1 - \sqrt{\lambda}\right)^2} \log\left(1 + \frac{\log\left(\left(1 - \sqrt{\lambda}\right)^{4/3}\right)}{\log\left(\frac{2\left(2 - 2\sqrt{\lambda}\right)^{2/3}}{3\sqrt[3]{\pi}\sqrt{\lambda}\epsilon^{2/3}}\right)}\right),$$

(25)

where the second term goes to zero as $\epsilon \to 0$, quoted in the main text as Eq. (11).

## C   DERIVATION FOR TRAINING LABEL NOISE

Here, we consider the addition of training label noise for the simplest setup, in which $d_{out} = 1$ so $S, T$ are vectors, and in the absence of WD. In this case, the noise addition is a scalar label noise variable $\delta_i \sim \mathcal{N}(0, \sigma_\delta^2)$. Training with label noise in the linear single-layer model gives the average training loss

$$\mathcal{L}_{\text{tr}} = \frac{1}{N_{\text{tr}}} \sum_{i=1}^{N_{\text{tr}}} \|(S - T)x_i + \delta_i\|^2 \simeq \text{Tr}(DD^T \Sigma_{\text{tr}}) + \sigma_\delta^2,$$

(26)

while the generalization loss (w/o noise) is unchanged

$$\mathcal{L}_{\text{gen}} = \frac{1}{N_{\text{gen}}} \sum_{i=1}^{N_{\text{gen}}} \|(S - T)x_i\|^2 \simeq \text{Tr}(DD^T)$$

(27)

. Since the label noise contribution does not multiply any of the weights, the gradient of the training loss with respect to $D$ is the one given in Eq. (3), so the training dynamics in the gradient flow limit remain unchanged

$$D(t) = e^{-4\eta_0 \Sigma_{\text{tr}}} D_0.$$

(28)

This implies that the training loss is bounded from below by $\sigma_\delta^2$, while the generalization loss can be driven to 0, indicating that even imperfect training in this model leads to perfect generalization.

In Fig. 6, we show the loss and accuracy evolutions for different values of $\sigma_\delta^2$. Clearly, for high noise variance, the training accuracy saturation is a poor metric for performance, while test loss and accuracy show perfect generalization for any noise level.

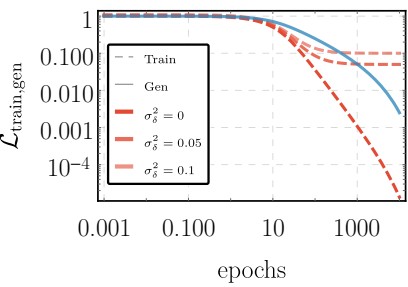 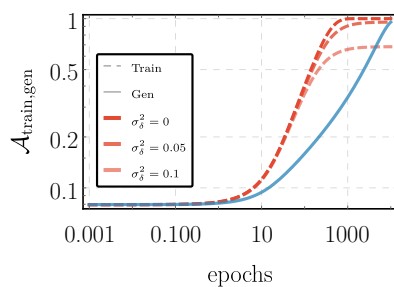

Figure 6: Training and generalization losses (**left**) and accuracies (**right**) for different noise label variance. In **shades of pink**, we show the solutions for Eq. (8) and Eq. (9). In **light blue**, we show the generalization loss/accuracy evolution which does not change as a function of label noise. Here, $\eta_0 = 0.01, \lambda = 0.9, d_{\text{out}} = 1., \epsilon = 0.01$.

## D    DERIVATION FOR $d_{\text{out}} > 1$

Here, we provide additional details on the derivation of Eq. (12). The starting point is the training and generalization loss functions, given by

$$\mathcal{L}_{\text{tr}} = \frac{1}{d_{\text{out}}} \text{Tr} \left[ D^T \Sigma_{\text{tr}} D \right], \qquad \mathcal{L}_{\text{gen}} = \frac{1}{d_{\text{out}}} \text{Tr} \left[ D^T \Sigma_{\text{gen}} D \right] = \frac{1}{d_{\text{out}}} \| D \|^2 . \tag{29}$$

where $S, T \in \mathbb{R}^{d_{\text{in}} \times d_{\text{out}}}$ are the student and teacher weight matrices, $\Sigma_{\text{tr}} \equiv \frac{1}{N_{\text{tr}}} \sum_{i=1}^{N_{\text{tr}}} x_i x_i^T$ is the empirical data covariance, or Gram matrix for the *training* set, and we define $D \equiv S - T$, the difference between the student and teacher matrices. $T$ and $S$ are drawn at initialization from normal distributions $S_0, T \sim \mathcal{N}(0, 1/(2 d_{\text{in}} d_{\text{out}}))$. We do not include biases in the student or teacher weight matrices, as they have no effect on centrally distributed data. The gradient descent equations in this instance are simply

$$D_{t+1} = \left( \mathbf{I} - \frac{2\eta}{d_{\text{out}}} \Sigma_{\text{tr}} \right) D_t, \tag{30}$$

where the only difference between the $d_{\text{out}} = 1$ case and the equation above is the rescaled learning rate $\eta \to \eta/d_{\text{out}}$ and the dimensions of $D_t$. Since the MP distribution is identical for each column of $D_t$, the results sum up and are identical to the $d_{\text{out}} = 1$ case for the losses, apart from a factor of $1/d_{\text{out}}$ and the learning rate rescaling, leading to Eq. (12).

## E    LOSS CALCULATIONS FOR DYNAMICS INCLUDING WEIGHT DECAY

Here, we provide the derivation for Eq. (14). We begin with the definitions of the loss function in the $d_{\text{out}} = 1$ case

$$\mathcal{L}_{\text{tr}} = D(t)^T \Sigma_{\text{tr}} D(t), \tag{31}$$

where $D(t) = S(t) - T$ is the difference between the student and the teacher vectors, $\Sigma_{\text{tr}} = \frac{1}{N_{\text{tr}}} \sum_{i=1}^{N_{\text{tr}}} x_i x_i^T$ is the training covariance matrix, and $\gamma \geq 0$ is the weight decay parameter. Using the gradient descent equation in the gradient flow limit, $\frac{\partial D}{\partial t} = -\eta \nabla_D \mathcal{L}$, we obtain from Eq. (3) that

$$\frac{\partial D}{\partial t} = -2\eta \left( \Sigma_{\text{tr}} + \frac{1}{2} \gamma I \right) D - \eta \gamma T. \tag{32}$$

Multiplying by the integration factor $e^{2\eta \left( \Sigma_{\text{tr}} + \frac{1}{2} \gamma I \right) t}$ and taking the integral, we arrive at

$$D(t) + \frac{1}{2} \gamma \left( \Sigma_{\text{tr}} + \frac{1}{2} \gamma I \right)^{-1} T = e^{-2\eta \left( \Sigma_{\text{tr}} + \frac{1}{2} \gamma I \right) t} \left[ D(0) + \frac{1}{2} \gamma \left( \Sigma_{\text{tr}} + \frac{1}{2} \gamma I \right)^{-1} T \right]. \tag{33}$$

We note that now the limiting value of $D(t \to \infty)$ is not zero, but rather $D_\infty = -\frac{1}{2} \gamma \left( \Sigma_{\text{tr}} + \frac{1}{2} \gamma I \right)^{-1} T$. Next, we wish to calculate $\mathcal{L}_{\text{tr}} = D(t)^T \Sigma_{\text{tr}} D(t)$ and $\tilde{\mathcal{L}}_{\text{gen}} =$

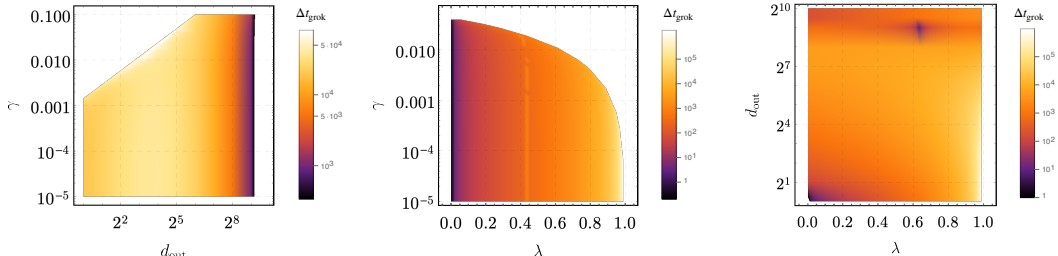

Figure 7: Grokking time phase diagrams. **Left:** A contour plot of the grokking time difference as a function of $\gamma, d_{\mathrm{out}}$. Shades of red indicate shorter grokking time, while blue tones indicate longer grokking time. White regions indicate no grokking, as generalization accuracy does not converge to 95%. **Center** and **Right:** Similar phase diagrams for the grokking time difference as a function of $\gamma, \lambda$ and $d_{\mathrm{out}}, \lambda$, respectively. The results of all three plots are obtained by numerically finding the grokking time, using the definition $\mathcal{A}(t^*) = 0.95$ and the analytic formulas quoted in the main text. The fixed parameters for these plots are $\eta_0 = 0.01, \epsilon = 10^{-3}$.

$D(t)^T \Sigma_{\mathrm{gen}} D(t)$, where we emphasize that in both cases $D(t)$ is given by Eq. (33) and depends on $\Sigma_{\mathrm{tr}}$. As described in the main text, it is a good approximation to set $\Sigma_{\mathrm{gen}}$ to be the identity matrix. For convenience, we will write both cases by $\mathcal{L}_{\mathrm{tr/gen}} = D(t)^T Q D(t)$, where $Q = \Sigma_{\mathrm{tr}}$ for the train and $Q = \boldsymbol{I}$ (the identity matrix) for the generalization.

We continue by diagonalizing $\Sigma_{\mathrm{tr}}$; we write $M = P^T \Sigma_{\mathrm{tr}} P$, where $M$ is a diagonal matrix whose eigenvalues follow the MP distribution. Hence, we obtain

$$\mathcal{L}_{\mathrm{tr/gen}} = \bar{D}(t)^T \bar{Q} \bar{D}(t), \tag{34}$$

where $\bar{Q} = M, I$ for the train, generalization correspondingly, and $\bar{D}(t)$ is given by

$$\bar{D}(t) = e^{-2\eta\left(M+\frac{1}{2}\gamma I\right)t} \left[ \bar{D}(0) + \frac{1}{2}\gamma \left(M + \frac{1}{2}\gamma I\right)^{-1} \bar{T} \right] - \frac{1}{2}\gamma \left(M + \frac{1}{2}\gamma I\right)^{-1} \bar{T}, \tag{35}$$

where $\bar{D}(t) = P^T D(t), \bar{T} = P^T T$. We notice now that the expression in Eq. (34) involves terms in the form of: $V^T f(M) W$ where $V, W$ are some vectors, and $f(M)$ is some function of the diagonal MP matrix. If $V, W$ are random vectors in a large dimension, we can approximate that

$$V^T f(M) W = \begin{cases} 0 & V \neq W, \\ |V|^2 \int f(u)p(u)du & V = W, \end{cases} \tag{36}$$

where $|V|$ is the norm of $V$, and $p(u)$ is the probability density function of the MP distribution. For example, in our case we will get that $D^T(0)f(M)T = -|T|^2 \int f(u)p(u)du$ (since $D(0) = S(0) - T$). All that is left now is to calculate the expression in Eq. (34) explicitly, using the approximation of Eq. (36). Doing this, at last we arrive into

$$\mathcal{L}_{\mathrm{tr/gen}} = d_{\mathrm{in}} \int \left( |S(0)|^2 e^{-4\eta\left(u+\frac{1}{2}\gamma\right)t} + |T|^2 \left( \frac{e^{-2\eta\left(u+\frac{1}{2}\gamma\right)t}u + \frac{1}{2}\gamma}{u + \frac{1}{2}\gamma} \right)^2 \right) q_{\mathrm{tr/gen}} p(u)du, \tag{37}$$

where $q_{\mathrm{tr}} = u$ and $q_{\mathrm{gen}} = 1$. By also setting the student initialization and teacher vector norms to $|S(0)|, |T| \simeq 1/\sqrt{2d_{\mathrm{in}}}$ (as done in the main text), we finally get

$$\mathcal{L}_{\mathrm{tr/gen}} = \frac{1}{2} \int \left( e^{-4\eta\left(u+\frac{1}{2}\gamma\right)t} + \left( \frac{e^{-2\eta\left(u+\frac{1}{2}\gamma\right)t}u + \frac{1}{2}\gamma}{u + \frac{1}{2}\gamma} \right)^2 \right) q_{\mathrm{tr/gen}} p(u)du. \tag{38}$$

## F   DERIVATION FOR THE 2-LAYER NETWORK

Here, we provide supplementary details on the derivation of Eq. (16). We consider the addition of a single hidden linear layer, where the teacher network function is $f(x) = (T^{(1)})^T (T^{(0)})^T x$, where

$T^{(0)} \in \mathbb{R}^{d_{\text{in}} \times d_h}$, $T^{(1)} \in \mathbb{R}^{d_h \times d_{\text{out}}}$ and $d_h$ is the width of the hidden layer. Similarly, the student network is defined by two matrices $S^{(0)}, S^{(1)}$. The empirical training loss over a sample set $\{x_i\}_{i=1}^N$ reads

$$\mathcal{L}_{\text{tr}} = \frac{1}{N_{\text{tr}} d_{\text{out}}} \sum_{i=1}^{N_{\text{tr}}} \left( (S^{(1)})^T (S^{(0)})^T x_i - (T^{(1)}))^T (T^{(0)}))^T x_i \right)^2 . \tag{39}$$

In this setup the weights are drawn at initialization from normal distributions $S_0^{(0)}, T^{(0)} \sim \mathcal{N}(0, 1/(2d_{\text{in}} d_h))$, $S_0^{(1)}, T^{(1)} \sim \mathcal{N}(0, 1/(2d_{\text{out}} d_h))$. Next, we define $T = T^{(0)} T^{(1)} \in \mathbb{R}^{d_{\text{in}} \times d_{\text{out}}}$ and derive the gradient flow equations for the system

$$\dot{S}_t^{(0)} = -\frac{2\eta_0}{d_{\text{out}}} \Sigma_{\text{tr}} \left( S_t^{(0)} S_t^{(1)} - T \right) (S_t^{(1)})^T, \quad \dot{S}_t^{(1)} = -\frac{2\eta_0}{d_{\text{out}}} (S_t^{(0)})^T \Sigma_{\text{tr}} \left( S_t^{(0)} S_t^{(1)} - T \right). \tag{40}$$

defining $D_t = S_t^{(0)} S_t^{(1)} - T$, and noting that $\dot{D}_t = S_t^{(0)} \dot{S}_t^{(1)} + \dot{S}_t^{(0)} S_t^{(1)}$, we arrive at the equations quoted in the main text

$$\dot{D}_t = -2\eta_0 \frac{h}{d_{\text{out}}^2} \Sigma_{\text{tr}} D, \qquad \dot{h}_t = -8\eta_0 (T + D)^T \Sigma_{\text{tr}} D. \tag{41}$$

Here, $h = Tr[H]/2 = \|S^{(0)}\|^2/2 + \|S^{(1)}\|^2/2$, where $H = \nabla_\theta^T \nabla_\theta \mathcal{L}_{\text{tr}}$ is the Hessian matrix and $\theta \equiv \{S^{(0)}, S^{(1)}\}$. Although Eq. (17) describes a set of coupled equations, we note that the solution for $h_t$ can be simplified when considering

the limit of small $\eta_0 \ll 1$, as we may ignore the time evolution and consider the trace (or kernel) as fixed to its initialization value, which is $h_0 \simeq 1/2$ for $d_h \gg d_{\text{out}}$. In that case the loss solutions are a simple modification to the ones given in the previous sections, with the replacement $\eta_0 \to \eta_0/(2d_{\text{out}}^2)$. Subsequently, the training/generalization performance metrics are

$$\mathcal{L}_{\text{tr/gen}}^{2-\text{layer}} = \|D_0\|^2 \mathcal{L}_{\text{tr/gen}}^{1-\text{layer}} \left( \frac{\eta_0}{2d_{\text{out}}^2}, \lambda, t \right), \quad \mathcal{A}_{\text{tr/gen}} = 1 - \frac{\Gamma \left( \frac{d_{\text{out}}}{2}, \frac{d_{\text{out}} \epsilon}{2\mathcal{L}_{\text{tr/gen}}} \right)}{\Gamma \left( \frac{d_{\text{out}}}{2} \right)}. \tag{42}$$

## G PRELIMINARY EVALUATION OF PREDICTION CONSISTENCY FOR REAL-WORLD SCENARIOS

Here, we begin to explore to what extent can our predictions for the linear models carry over to real-world settings. As an example, we consider the classical modular arithmetic task, discussed in Power et al. (2022), and later in Gromov (2023).

We borrow the notations and construction from Gromov (2023), using the same architecture and definition of the problem. In this setting, grokking takes place and learned features can be understood analytically. We consider a two-layer MLP network without biases, given by

$$h_k^{(1)}(x) = \sqrt{\frac{1}{d_{\text{in}}}} \sum_{j=1}^{d_{\text{in}}} W_{kj}^{(1)} x_j, \quad z_i^{(1)}(x) = \phi(h_i^{(1)}(x)), \quad h_q^{(2)}(x) = \frac{1}{d_h} \sum_{k=1}^{d_h} W_{qk}^{(2)} z_k^{(1)}(x), \tag{43}$$

where $d_h$ is the width of the hidden layer, $d_{\text{in}}$ is the input dimension, and $\phi$ is an activation function, which we take to be the preactivation squared $\phi(h) = h^2$. At initialization the weights are sampled from the standard normal distribution $W^{(1)}, W^{(2)} \sim \mathcal{N}(0, 1)$. In Eq. (43), we have chosen to follow the mean-field parametrization Mei et al. (2018)[1].

Given this architecture, we then set up the task of modular arithmetic as a classification problem. The network is tasked with learning

$$f(n, m) = (n + m) \mod p \tag{44}$$

where $m, n, p$ are integers. To this end, we fix $p$ and consider additive over $\mathbb{Z}_p$. Each input integer is encoded as a one-hot vector. The output integer is also encoded as a one-hot vector. For the task of

---

[1] In the limit of infinite width, the meanfield parametrization allows for feature learning.

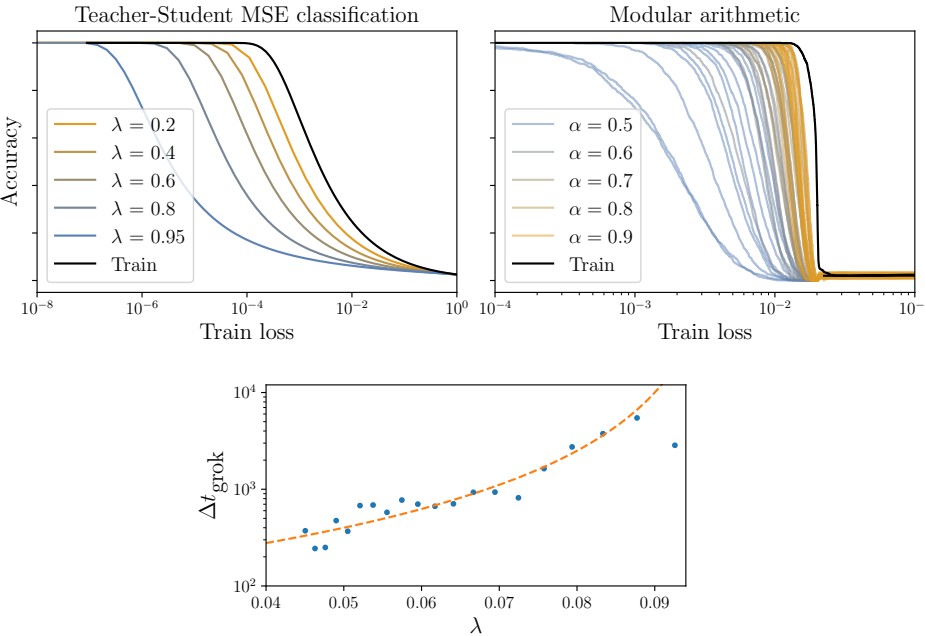

Figure 8: Grokking dependence on $\lambda$ for a modular addition task compared against the Teacher-Student task given in the main text. **Top row:** In the *left* panel, we show the accuracy as a function of training loss for the Teacher-Student task, while in the *right* panel, we show the same metric for the modular addition task. **Different shades** indicate a different $\lambda$ value for the Teacher-Student task, or different $\alpha$ values for modular arithmetic. **Blue** indicates the smallest number of training samples, leading to longer grokking time, while **yellow** curves correspond to more training samples, therefore shorter grokking time differences. Here, we take $p = 48$ and width $d_h = 400$. **Bottom panel:** Grokking time difference as a function of $\lambda = 2/(\alpha p)$ for modular addition. The dashed line shows a $(\lambda - \lambda^*)^{-2}$ dependence.

learning bivariate functions over $\mathbb{Z}_p$ the input dimension is $d_{\text{in}} = 2p$, the output dimension is $p$, the total number of points in the dataset is $p^2$, while the model in Eq. (43) has $3d_h p$ parameters. Finally, we split the dataset $\mathcal{D}$ into train $\mathcal{D}_{\text{train}}$ and test $\mathcal{D}_{\text{test}}$ subsets, and furnish this setup with the MSE loss function.

It was observed in Gromov (2023) that grokking in this setup depends on multiple factors, including the training sample fraction $\alpha = |\mathcal{D}_{\text{tr}}|/|\mathcal{D}| = N_{\text{tr}}/N_{\text{gen}} = N_{\text{tr}}/p^2$. In our parameterization, we can study the behavior of the grokking time $\Delta t_{\text{grok}}$, defined in Eq. (11) as a function of $\lambda = d_{\text{in}}/N_{\text{tr}} = 2p/(\alpha p^2) = 2/(\alpha p)$, by changing the number of training samples, keeping the parameter $p$ fixed, and note whether there is any hint of the grokking time being a function of the loss alone.

In the top row of Fig. 8, we compare the behavior of the accuracies as a function of training loss with different training set sizes for the vanilla linear teacher-student model with $d_{\text{out}} = 1$, and for the modular addition task with $p = 48$[2]. We use the natural parameterization for the training set size in both cases, i.e., $\lambda$ in the teacher-student model, and $\alpha$, the fraction of training samples, for modular addition[3]. We further show on the bottom row of Fig. 8, the grokking time difference as a function of $\lambda$. We find that there is some indication that the grokking time difference in modular addition depends on $\lambda$ as $\sim (\lambda - c)^{-2}$, which is reminiscent of the linear model. Additionally, comparing the accuracy/training loss curves, one finds that as grokking begins, the linear and the modular losses and accuracies follow similar trends, indicating that perhaps the linear model predictions can be extended to become universal, under the right reparameterization of the problem.

---

[2]In order to expedite convergence we used a learning rate of $\eta = 500$. This is in principle "large", but does not affect the standard grokking behavior observed in Gromov (2023), where the learning rate was set to $\eta = 100$.

[3]Here, $\alpha$ ranges between 0.5 and 0.9, since below $\alpha = 0.5$, the network cannot generalize.

