# OpenReview forum: "Grokking in Linear Estimators -- A Solvable Model that Groks without Understanding"
_ICLR.cc/2024/Conference — ICLR 2024 poster_

### Official Review · Reviewer_VxMM · 2023-10-23

**Soundness:** 3 good
**Presentation:** 3 good
**Contribution:** 2 fair
**Rating:** 3
**Confidence:** 4

**Summary:**

This manuscript studies a linear teacher-student model trained with Gradient flow and MSE loss. The authors solve the dynamical training equation which yields to an exponential decay of the initial error with a spectrum of exponents related to the training data covariance matrix. When the input dimension is larger than the number of data-points, this matrix has zero eigenvalues and hence target modes that cannot be learned. Moreover, using Gaussian iid data, this matrix is Wishart, thereby allowing the authors to estimate where the bulk of non-zero spectral modes resides. Following this they find that the asymptotic dependence of the training loss is roughly exponential with a time scale depending on the lower bulk spectrum of the Wishard matrix. In this asymptotic regime, the test loss behaves as the train loss up to a multiplicative factor larger than 1. Hence the time at which the training loss drops below some small threshold (t_{*,train}) is smaller than the time it takes the test loss to do so (t_{*,test}). This delay between t_{*,test} and t_{*,train} is then viewed as a form of Grokking. The authors proceed with extending their results to several other scenarios. Such as networks with vector outputs, weight decay, and two-layer network in the linear/NTK regime.

**Strengths:**

The authors study a timely topic.

They lay bare some issues with how Grokking is perceived.

The manuscript is clearly written and easy to follow.

**Weaknesses:**

I feel there are two main issues with the current work.

1. Is this Grokking?

Grokking is a somewhat ill-defined phenomenon. However, like many grey area situations, the fact that it is difficult to draw one sharp reason doesn't mean there cannot be a sharp feeling that something does fit. Let me then rationalize several reasons for why I don't think this is grokking:
A. It is too simple. If this model Groks, everything Groks.
B. There aren't any feature learning effects (see for instance Gromov (2023) or Misra (2022)) or beyond NTK effects.
C. There is only one time scale here, which is (1-\sqrt{\Lambda})^{-2}, governing both test and train loss. Hence, looking at the train and validation loss, a practitioner would not be encouraged to do any early stopping here. In contrast, in various Grokking models, there is an over-fitting regime prior to Grokking which can mislead practitioners.

2. Technical aspects seem quite close to previous results in the literature.

The training dynamic the authors find is closely related to the e^{-\Theta t} \Delta obtained in the NTK work. In fact, up to a standard shift of the inner and outer indices, their data covariance matrix is Theta here. In particular, both have the same non-zero spectrum. I'm sure there are much earlier works which solve this for a linear network.

Similar factors relating train and test performances, however at infinite time, were also found in Adlahm and Pennington 2020 and https://arxiv.org/abs/2006.09796. Given the expected exponential decay of the discrepancy in linear models, the fact this ratio holds also in the asymptotic dynamics seems expected. At any rate, It'll be good to compare and disentangle the factors the authors find from those.

**Questions:**

The work is clearly written. I don't have any questions.

---

> ### Author Response · Authors · 2023-11-23
>
> Thank you for your thoughtful review and constructing comments and questions.
> We are glad that you acknowledge the main goal of our work, which is to call for a more serious discussion on what grokking is, and properly define it by multiple metrics that must converge to a clear conclusion.
>
> Before addressing the weaknesses raised in the review, we must first provide a small correction to how our results were perceived. Most of our analysis was done in the regime $d_{in}/N_{tr}<1$, meaning that there are more training samples than input dimensions. In this case, there are no zero eigenvalues, and the difference between training and generalization loss comes from the mismatch between the covariance matrices for the nonzero eigenvalues.
>
> Next, we address the weaknesses in the order in which they were raised.
>
> 1) **Is this grokking?**
>
> This is indeed an important point that also came up in the reviews of other referees. We agree, and it is in fact one of the main conclusions of our work. Namely: we study a model which does not exhibit any feature learning, no special weight structure or non-monotonicity in its generalization loss evolution, and can still be considered to "Grok" if one uses accuracy as the only metric (and this specific case was actually descried as grokking in the Omnigrok paper [Liu 2023]).  Truly, if this nearly trivial model Groks, then any model can Grok.
>
> We believe that this aspect of delayed generalization should be separated from representation learning grokking which is present in other systems, such as modular arithmetic. However, we believe that some aspects of the proposed mechanism -- namely that the accuracy is simply a step-like-function of the loss that depends on the data covariance -- might be quite general. We provide some preliminary evidence in the new appendix G.
>
> We agree with the suggestion of tracking both generalization loss and accuracy to define grokking, but even in this scenario there may be other factors, such as optimizer dependence that can induce delayed generalization on the loss, without any deep change in learning behavior.
>
> We completely rewrote tyhe discussion to better emphasize this point
>
> 2) **Similarity to previous works**
>
> While we recognize some similarities to previous work, we believe our study makes several novel contributions:
>
> We provide a closed form solution to the dynamics of training for linear models, rephrasing previous asymptotic analyses in simple Random Matrix Thoery language. This allows us to characterize training dynamics over the full training process and isolate the contribution of different aspects.
>
> By relating training loss directly to the spectrum of the data covariance matrix, we suggest that perhaps some of our results can be extended beyond the linear regime, as shown by the 2-layer $\tanh$ example, as well as a new appendix (Appendix G.) in which we examine the dependence of the grokking time on $\lambda$ for a modular arithmetic task.
>
> 3) **Finite time analysis vs previous works:**
> We agree that previous works have examined the exponential decay of the discrepancy in linear models, and therefore the novelty in our work is relating that same discrepancy with the phenomena of grokking. We added some of your suggestions to the bibliography.
>
> We hope with these changes the main message of the paper is clearer, while the broader context is more readily accessible, and that upon review of the revised version you find our work suitable to acceptance.

---

### Official Review · Reviewer_Eifc · 2023-10-26

**Soundness:** 3 good
**Presentation:** 2 fair
**Contribution:** 3 good
**Rating:** 8
**Confidence:** 3

**Summary:**

This paper uncovers a surprising fact that linear models can manifest the 'grokking' phenomenon within a teacher-student framework. The author qualitatively predicts the precise dependencies of grokking time on input/output dimensions, sample size, regularization (weight decay), and initialization. Remarkably, these predictions hold true for more intricate models in specific settings. The paper also includes a comprehensive set of empirical validations to support the theoretical analysis.

**Strengths:**

The paper is notably well-structured and written. Its strengths can be summarized as follows:

- The paper offers a substantial novelty value, as it explores the grokking phenomenon through a theoretical lens, an area with limited prior analysis. The surprising discovery of grokking in certain linear model scenarios (even without weight decay) adds a unique perspective to the literature. This novel approach contributes significantly to our understanding of grokking from the first principle.
- The authors provide comprehensive theoretical predictions for their proposed linear model. They not only establish the asymptotic order of grokking time but also consider various related factors. Also, the paper extends its analysis to two-layer networks, both with and without nonlinearity. To further strengthen their claims, the authors conduct an array of extensive experiments that effectively illustrate and validate the theoretical findings.

**Weaknesses:**

Though the paper is of good quality, I believe the paper would improve in the following aspects:
- The notations are quite dense even for the simplest model at first glance. The paper would benefit from having some explicit definition or intro to the model and all the notations.
-  The derivation of this paper is concrete. However, all the calculations are in the asymptotic manner. It would be better to see the results be displayed as some rigorous theorem in some limiting case.

**Questions:**

In this linear model, the grokking phenomenon is due to the gap of convergence speed between training and generalization loss. Can it also imply the difference in two-layer networks settings (or in NTK regime), or they have a different mechanism?

---

> ### Author Response · Authors · 2023-11-23
>
> Thank you for your thoughtful review and positive appraisal of our work. We are glad that you found our analysis of grokking through a theoretical lens in linear models and neural networks to be a novel contribution that substantially advances understanding. You also highlighted how our comprehensive theoretical predictions and extensive empirical validations effectively supported our claims. These are certainly important strengths of the work.
>
> At the same time, you raise valid points about areas we can improve.
>
> 1) **Notations:**
> We agree. We edited the manuscript to make the notations clearer and easier to follow. Thank you for your comments.
>
> 2) **Theorems:**
> Following your comments, we have added a short introduction to the model at hand, clarifying the notations we used. We have also added references to certain theorems which substantiate our asymptotic results in more limiting cases.
>
> 3) **NTK limit:**
> Thank you for the question. We believe that the answer is that in the lazy regime, i.e. when the kernel remains fixed at initialization $h_t\simeq h_0$ our analysis will hold for NTK, however any deviation from the strict NTK limit may generate dynamics that differ for our results, in any truly nonlinear settings. This implies that some of our predictions regarding the scaling of grokking time with certain parameters, for instance, $d_{in}$ and $N_{training}$ should hold, at least to some extent, until breaking away fully from linear dynamics.

---

### Official Review · Reviewer_minU · 2023-10-30

**Soundness:** 4 excellent
**Presentation:** 2 fair
**Contribution:** 3 good
**Rating:** 5
**Confidence:** 4

**Summary:**

This paper proposes a toy model for grokking, a phenomenon where training accuracy rises much before test accuracy. The proposed model is a linear target function on Gaussian data distribution. A student model attempts to estimate parameters with a matching linear model. The authors use random matrix theory (specifically the eigenvalue distribution of the Wishart ensemble) to derive the dynamics of training and test losses under gradient flow. While both train and test losses decrease smoothly in terms of MSE, when plotting a hard classification accuracy, this toy model exhibits a separation of timescales between training accuracy saturation and test accuracy saturation. The authors illustrate that this is entirely due to the difference between the feature covariance matrix on the training distribution and on the test distribution. These results, taken together provide a possible deflationary explanation for grokking, which can occur merely as an artifact of the choice of metric in very simple models, rather than due to a deeper reason related to learning generalizing features at late training time. The authors show that their results can be extended to weight decay, multiple outputs, training a linear neural network with multiple layers, and training deep nonlinear networks in the lazy/kernel regime.

**Strengths:**

This paper has a nice analytically solvable theory of training dynamics which reproduces grokking-like learning curves. The paper uses a basic result in random matrix theory, namely the limiting spectral density of the Marcheko Pastur law to derive the train and test loss dynamics. Due to the model’s simplicity the authors can derive nice asymptotic expressions for the loss at late time and use these to derive a grokking timescale. They show that their theory is predictive in experiments on reasonably large features and datasets sizes.

**Weaknesses:**

While this paper provides an interesting and exactly solvable toy model of grokking, it has some major defects which must be addressed before I can support acceptance.

First, the paper lacks a proper comparison to literature on gradient flow dynamics in linear models (see below). In addition the citations are often incorrect or correspond to nonexistent papers. We outline these issues below in section titled Related Work Issues.

Next, the comparison of the grokking observed in this model and grokking observed in real networks is still unclear. Do the authors think that the proposed statistical effect is the phenomenon at play in grokking in real networks? Below I list some observed phenomena associated with grokking that this theory does not quite capture

1.  Many works on grokking report development of specialized weight structures in deep networks temporally coincident with the improved generalization error. I suspect this form of grokking would correspond to deviation from a linear model which would be outside the scope of explanation in this work. If the authors are not claiming this is a theory of grokking as it is observed in its original settings (like modular arithmetic in transformers), in what sense is this an explanation of grokking?
2. Another example of a difference between observed dynamics and the present study is that weight decay does not induce more extreme grokking which was reported in Liu et al 2023.
3. Sometimes in grokking experiments with deep networks, the test loss (MSE) even increases before later decreasing to its final value like in Davies et al 2023 Figure 1. It is unclear if this can happen in the linear model where the errors in each non-null eigendirections decrease exponentially.

On the other hand, it could be that the primary point of the paper is motivating the need for more careful research on grokking in the future. For example, what artifacts due to choice metric should experimenters be wary of? How should we distinguish “improvement in understanding” vs mere statistical noise? Commenting further on this distinction and comparison to phenomena observed in prior works on grokking would greatly improve the paper.

**Issues with Related Work**

1. The authors miss several important works on gradient flow with linear models. Advani & Saxe 2020 (https://www.sciencedirect.com/science/article/pii/S0893608020303117) derive the training and test errors dynamics for the model considered in this paper. Omitting this citation does not do justice to their relevant contribution. In addition, Saxe et al 2013 (https://arxiv.org/abs/1312.6120) derive learning curves for deep linear networks with fixed dataset.
2. Several other papers which derive asymptotic performance of linear models trained with Gaussian data in the proportional regime with gradient flow.
   (a) Mignacco et al 2022 (https://proceedings.neurips.cc/paper/2020/hash/6c81c83c4bd0b58850495f603ab45a93-Abstract.html), Mignacco et al 2022 (https://iopscience.iop.org/article/10.1088/1742-5468/ac841d/meta),
   (b) Paquette et al 2022 (https://arxiv.org/abs/2205.07069)
3. The authors also miss the work of Gromov (https://arxiv.org/abs/2301.02679) on the dynamics of grokking in modular arithmetic without regularization.
4. Several cited papers do not even seem to exist! These include [2] E. Bodin and N. Macris. Dynamics of generalization in learning with gradient descent for piecewise linear neural networks [4] A. Crisanti and H. Sompolinsky. Dynamics of learning in deep linear neural networks: A mean-field approach   [7] S. Goldt, M. M’ezard, F. Krzakala, and L. Zdeborov’a. Modelling the infinite width limit of neural networks with mean field theory

Please fix these issues.

**Questions:**

**Questions/Comments**

1. Figure 5 Bottom row: Is the reason that the dynamics for the tanh networks are well approximated by linear two layer dynamics because the activation function is linearizable around $\phi(h) \sim h$? What if you used a different activation function, for instance, increasing the “gain” of the tanh nonlinearity like $\tanh(g * h)$ for $g > 1$? For large enough $g$, would we expect the correspondence to the linearized dynamics to break?
2. Equation 7 is using approximation signs instead of equality. However, I suspect that if one considered the average case error over random teachers where $T \sim \mathcal N(0,I)$ then the equation given is exact. Is there any reason to not focus on averaging over random teachers since the data is isotropic?
3. The authors claim that deep networks in the kernel regime can be described by the same equations except redefinition of the kernel. I believe that the kernel would not only change the eigenvalue spectrum but would also make the train and test dynamics dependent on the decomposition of the teacher T in the eigenbasis of the kernel. This would therefore make the result much less trivial than what is provided here since the test error is not merely a functional of the spectral density.

---

> ### Author Response · Authors · 2023-11-23
>
> Thank you for your careful reading and thoughtful comments. We are glad that you believe we presented a sound and comprehensive analysis of our simple model which captures some aspects of grokking.
>
> In light of the weaknesses and questions brought up, we have made a number of modifications to the text, as detailed below.
>
> * **Comparison to gradient flow in linear models literature and related works:**
>     We apologize for the errors found in the bibliography and the related work section. Particularly, the work of Gromov was one of the motivations for our work and was somehow unintentionally missed. We thank the reviewer for bringing these highly relevant works on linear models (for instance Saxe et al.) and on gradient flow to our attention. they are now incorporated into the main text.
>
> * **Extension of the results to more realistic setups:**
> We fully agree that our model does not capture all aspects of grokking. As we wrote in our response to other referees, this is actually a main point of our paper: in some cases grokking is just an artifact of the accuracy measure, and does not result from an ``interesting`` thing that goes on in the underlying dynamics. This was stated explicitly in the abstract and the text. We rewrote the discussion to stress this point.
>
> We agree with your ultimate conclusion: "...the primary point of the paper is motivating the need for more careful research on grokking in the future. For example, what artifacts due to choice metric should experimenters be wary of? How should we distinguish “improvement in understanding” vs mere statistical noise?..." . This was a main goal of this work. Since grokking is ill-defined, even a linear model which does not "understand" anything deep, can still "grok" because of an artifact.
>
> That said, it is still possible that at least some aspects of our analysis would extend to realistic setups, for instance the grokking time dependence on $d/N$, at least for some models. That is, at least some of what is currently considered "grokking" can be explained using our formalism. We begin to address this point in the new appendix G, but this is clearly an interesting point for future research.
>
> We stress in the discussion that our work is not intended to be a "theory of grokking" in general.
>
> Regarding the specific aspects of grokking that are missing in our model:
>
> 1) **Specialized weight structure:**
> Since the linear model does not exhibit any type of feature learning or gains any deep insight during training, there is no specialized structure for the weights. Indeed, this is a deviation from grokking in modular arithmetic.
>
> 2) **The effect of weight decay:**
> In the linear model, the effect of WD can be analytically understood, and in fact the reason for the difference is that they explore WD values in our "gray" regime, as our model cannot generalize in that parameter space while their setup can.
> We have revised our manuscript to explain this point.
>
> 3) **Loss increase prior to grokking:**
> Indeed, this is also a feature that does not exist in the linear model, as all the quantities in the model change monotonically with one another, and so unless one appeals to a catapult scale learning rate in the two-layer example, this cannot happen. However, our preliminary results (in the new appendix G) hint that this non-monotonicity might not be crucial for grokking (see figure 8, right panel, where the non monotonicity is present).
>
> **Questions/Comments:**
>
> 1) **$\tanh$ gain:**
> Yes, we believe that the near linear behavior of $\tanh$ is the reason for the matching results, since the analysis would not hold for a general activation function. For instance, we find empirically, that using ReLU activations completely deviates from the simple prediction of $h_t\simeq h_0$. Extending $g>1$ does break the predictions as well, above a certain threshold. Due to time constraints we were not able to incorporate a discussion in our work.
>
> 2) **Averaging over teachers:**
> It is probably true that the average over teachers would turn approximations to equalities. We show a weaker analytical claim, but a stronger empirical verification: our model give effectively perfect predictions for the dynamics of *individual trajectories*, and not average predicitons on ensembles.
>
>
>
> 3) **Teacher decomposition on the kernel**
> This question requires more substantial analysis - while it is true that the decomposing the teacher in the eigenbasis of the kernel would imply dependence on the eigenfunctions themselves, the question of how these functions evolve during training would likely determine their importance. In the infinite width limit, since the kernel is fixed this effect should not exist, but we agree that there are some non-trivial calculations to be made.

---

### Official Review · Reviewer_wJux · 2023-11-01

**Soundness:** 3 good
**Presentation:** 3 good
**Contribution:** 2 fair
**Rating:** 6
**Confidence:** 4

**Summary:**

This paper studies delayed generalization in a linear student-teacher setup with diagonal-covariance Gaussian inputs. The authors also offer some extensions to non-linear 2-layer networks (still in the Gaussian setting). Extensive analytical derivations in the gradient flow regime are shown to match numerical simulations closely.

**Strengths:**

- The paper is very well-written and easy to follow.
- Corroborates prior observations on the effect of dataset size on Grokking and distills the delayed generalization behavior down to a very simple toy setup.
- Theoretically and empirically, the toy setting is well explained and studied thoroughly, from a simple 1-D linear setup to various extensions, including a non-linear 2-layer setting.

**Weaknesses:**

- My biggest worry about the paper is that it does not deviate from the toy setting proposed. While the results are good, and the match between analytical results and empirical behavior is close, the entire investigation leaves out realistic applications and the original setting in which the phenomenon was first observed. Furthermore, it seems like the theory does not cover some phenomena associated with Grokking, and it might just be too far from the original setting to yield practical results.
The submission would be substantially stronger with more connections to realistic settings or insights that can transfer across, e.g., tasks, datasets, models, etc.

Overall, I would tend to recommend acceptance because the thorough investigation and the interesting insights on a simplified toy setting outweigh the limited scope.

**Questions:**

- Delayed generalization in many settings can occur after the generalization loss **increases**, whereas the toy setting presented seems to imply monotonic behavior. Could this mean that the phenomenon studied here is qualitatively different from Grokking in the original setting? There is no fixed and precise definition of Grokking, so this is not a big issue, but I think a distinction might be helpful in understanding feature learning dynamics.
- The effect of label noise does not seem to be completely captured by the theory. I have seen scenarios where the label noise is “corrected” at Grokking time, still reaching 100% accuracy. Specifically, one can take a *Grokking on modular addition* setup and assign random labels at a certain rate. Generalization accuracy can still reach 100% eventually. Do you have any idea why that’s the case? Is there a way for the toy model to accommodate such "self-correcting" behavior?

**Nit:**
The figures are nice but can be difficult to parse. In particular, I think it would be easier to understand figures where the theoretical predictions are differentiated for the training and generalization curves (e.g., dashed lines for train, solid for generalization with empirical data shown as points with different markers for train/gen).

---

> ### Author Response · Authors · 2023-11-23
>
> Thank you for your positive feedback on our paper. We appreciate you highlighting its clarity and its thorough explanations of the theoretical analysis and empirical results through a tractable model. Below we address the weaknesses and questions raised, in hopes of strengthening the message of our work.
>
>
> **Weaknesses:**
>
> We agree with your concerns that our conclusions might not extend beyond the linear model, and that our model does not capture some of the phenomena associated with grokking, e.g. a non monotonic loss. A main goal of this work was to highlight that the grokking phenomenon is not sufficiently well defined, and may contain multiple aspects, some of which can be captured (or mimcked?) by a nearly trivial model which shows no feature learning at all. This is indeed the main point we are trying to make, as we hint in the title of the manuscript.
>
> It should also be noted the model was not introduced in this work - we are analyzing a grokking scenario reported in the Omnigrok paper, and show that grokking in this nonlinear case can be understood in terms of a linear model (as we show in the two layer $\tanh$ example). That is, at least some of our analytical predictions might extend to nonlinear models. For instance, the change in grokking time as a function of $d_{in}/N_{training}$ is expected to hold in some form for more realistic setups as well. We have added experimental evidence in the revised manuscript that indicate that the grokking time scaling behavior extends for deeper nonlinear networks, for the task of learning modular arithmetic. We hope this change will provide a sufficiently broaden the scope of our work.
>
> Questions:
>
> 1) **Delayed generalization vs. grokking:**
> This is an excellent question, which is related to the previous point. Indeed, we agree that if grokking is defined by representation learning - hence requiring a dramatic change both in loss and in accuracy, this model does not grok but just pretends to. For instance, grokking in modular arithmetic clearly displays an ascent of the test loss while training accuracy saturates, and a drop of the test loss when the test accuracy increases from 0 to $100\%$. One of our goals in this work is to precisely separate the effects which depend only on the accuracy measure, from true representation learning. We also have preliminary results which may indicate that even the loss behavior is not necessarily indicative of "representation grokking", which we plan to present in future work.
>
>
> 2) **Self correction and label noise:**
> It is true that label noise was only commented upon in a paragraph in the original manuscript, but we can easily explain its effect here. Assume training in the most basic setup, in which $d_{out}=1$ so $S,T$ are vectors and there is a scalar label noise variable $n\sim\mathcal{N}(0,\sigma_n^2)$. Training with label noise in our model can be shown to give the average training loss $\mathcal{L}_{tr} =Tr(D_0 D_0^T  e^{-4\eta_0 \Sigma_t } \Sigma_t )+\sigma^2_n$,
> while the generalization loss (w/o noise) is unchanged, thus adding noise to test labels will add a similar additive term.
> Therefore, the training dynamics do not change except for an additive constant. It is then possible to tune $n$ and the threshold parameter $\epsilon$, such that the noise can cause induce grokking precisely when test accuracy saturates. There is no sense of self correction here, just a tuning between noise and the accuracy metric, so it is unclear if this is the same mechanism at work in modular addition.
>  We thank you for highlighting this point and we included an appendix explaining this.
>
> **Figures:**
> We thank you for this comment, and we have reworked our figures to be more clearer and more legible.

---

### Meta-Review · Area_Chair_5at1 · 2023-12-11

**Metareview:**

The authors present a linear model of the "Omnigrok" phenomenon (Liu et al 2022), a generalization of the original "Grokking" phenomenon beyond the algorithmic tasks. By simplifying to a linear setting, tools from random matrix theory can be used to analyze and provide insight into the phenomenon. Some reviewers had concerns that the definition of grokking used in the paper strayed too far from the original, which was focused on how LLMs can rapidly switch from memorizing solutions to an algorithmic problem to correctly learning the true solution. However, this seems to be an issue with the Omnigrok paper rather than the present submission, which merely simplifies the Omnigrok setting to the linear case. Some reviewers also raised issues with the presentation of the paper, which had some mistakes in the citations, but it appears that these issues have been fixed in the current submission. It is unfortunate that the reviewers did not engage during the discussion period, but from what I can see, the major issues raised by the reviewers have been adequately addressed. It would be especially interesting if the analytic approach taken in this paper could provide insight into the original grokking phenomenon on algorithmic tasks. Even so, I recommend acceptance.

**Justification For Why Not Higher Score:**

This was addressed in the metareview.

**Justification For Why Not Lower Score:**

This was addressed in the metareview.

---

### Decision · Program_Chairs · 2024-01-16

Accept (poster)